# Rational design of Al$_2$O$_3$/2D perovskite heterostructure dielectric for high performance MoS$_2$ phototransistors

Jiayang Jiang[1], Xuming Zou [1✉], Yawei Lv[1], Yuan Liu [1], Weiting Xu[1], Quanyang Tao[1], Yang Chai[3] & Lei Liao[1,2✉]

Two-dimensional (2D) Ruddlesden-Popper perovskites are currently drawing significant attention as highly-stable photoactive materials for optoelectronic applications. However, the insulating nature of organic ammonium layers in 2D perovskites results in poor charge transport and limited performance. Here, we demonstrate that Al$_2$O$_3$/2D perovskite heterostructure can be utilized as photoactive dielectric for high-performance MoS$_2$ phototransistors. The type-II band alignment in 2D perovskites facilitates effective spatial separation of photo-generated carriers, thus achieving ultrahigh photoresponsivity of >10$^8$ A/ W at 457 nm and >10$^6$ A/W at 1064 nm. Meanwhile, the hysteresis loops induced by ionic migration in perovskite and charge trapping in Al$_2$O$_3$ can neutralize with each other, leading to low-voltage phototransistors with negligible hysteresis and improved bias stress stability. More importantly, the recombination of photo-generated carriers in 2D perovskites depends on the external biasing field. With an appropriate gate bias, the devices exhibit wavelength-dependent constant photoresponsivity of 10$^3$–10$^8$ A/W regardless of incident light intensity.

[1] Key Laboratory for Micro/Nano Optoelectronic Devices of Ministry of Education & Hunan Provincial Key Laboratory of Low-Dimensional Structural Physics and Devices, School of Physics and Electronics, Hunan University, Changsha 410082, China. [2] State Key Laboratory for Chemo/Biosensing and Chemometrics, School of Physics and Electronics, Hunan University, Changsha 410082, China. [3] Department of Applied Physics, The Hong Kong Polytechnic University, Hong Kong 999077, China. ✉email: zouxuming@hnu.edu.cn; liaolei@whu.edu.cn

Photodetectors, which can capture light signals and convert them into electric outputs, are fundamental devices for modern imaging and communication applications[1–3]. With the development of photoactive materials and integration technologies, the maturity in functional diversity and production scale of photodetectors has reached a high level. However, the need for high-performance photodetection in terms of photosensitivity, linear response, spectrum coverage, flexibility, as well as the feasibility of integration with complementary metal oxide semiconductor also become more eminent. In this regard, organic–inorganic hybrid perovskites (OHPs) have recently emerged as appealing photoactive materials for optoelectronics because they offer multitude of extraordinary properties, such as high light absorption coefficient, low exciton binding energy, long electron–hole diffusion length, and especially cost-effective solution-based processes[4–6]. Unfortunately, the photodetectors based on single perovskite generally exhibit low photoresponsivity ($R$) due to their extremely low carrier mobility and the lack of an effective photoconductive gain ($G$) mechanism to produce multiple charge carriers upon one incident photon[7–9].

Building of hybrid phototransistors composed of perovskite-decorated two-dimensional (2D) materials, such as graphene[10], MoS$_2$[11,12], and black phosphorus[13], has been demonstrated to be an effective route to improve device performance. The operation mechanism relies primarily on efficient light absorption by perovskite and photo-generated electron-hole pairs separation at 2D materials/perovskite interface, which can prolong the carriers lifetime ($\tau_{\text{lifetime}}$), and thus improving $G$ value of the system. Nevertheless, the inherent instability induced by low formation energy of 3D structured perovskites limits their further development for commercialization. Recently, ⟨100⟩-oriented family of 2D Ruddlesden-Popper perovskites $(RNH_3)_2(A)_{n-1}M_nX_{3n+1}$ have been developed[5,14,15], wherein $RNH_3$ represents the organic spacer, $n$ is the number of 2D perovskite layers, and the small cation A, divalent metal cation M, and halide anion X construct the perovskite framework. Due to the hydrophobicity of organic spacer, $(RNH_3)_2(A)_{n-1}M_nX_{3n+1}$ perovskites exhibit superior stability. However, existing hybrid phototransistors are constructed by simple heterojunction stacking approach, which inevitably introduces undesirable doping into the underlaying 2D materials, leading to the reduced carrier mobility together with the loss of gate control[10–13]. At the same time, the insufficient spatial separation of photo-generated carriers hampers the optimization of photogain. Moreover, ionic migration induced hysteresis issue in perovskites is also a major impediment for the stable operation of perovskite-based photodetectors[16–18].

In this work, we design sensitive multilayer MoS$_2$ phototransistors based on an Al$_2$O$_3$/2D perovskite heterostructure dielectric. The hysteresis loop induced by ionic migration in 2D perovskite is opposite to that induced by charge trapping in Al$_2$O$_3$. By modulating the activation energy ($E_\alpha$) for ionic migration in 2D perovskites, the two negative effects of ionic migration in perovskite and charge trapping in Al$_2$O$_3$ may neutralize with each other. Thus, this heterostructure dielectric can effectively eliminate the hysteresis issue and significantly improves device reliability. At the same time, the 2D perovskite component serves as a high-efficiency photosensitizer to gate the MoS$_2$ channel. The type-II band alignment along the direction perpendicular to the substrate caused by the ordered distribution of 2D perovskite phases prompts the efficient spatial separation of photo-generated carriers, resulting in ultrahigh responsivity ranging from visible to near-infrared (NIR) region (>10$^8$ A/W at 457 nm and >10$^6$ A/W at 1064 nm). Moreover, owing to the external biasing field-dependent photo-generated carriers recombination in 2D perovskite, the device exhibits a constant responsivity of $2.4 \times 10^5$ A/W, a corresponding specific detectivity

of $5.5 \times 10^{12}$ Jones, and a large linear dynamic range (LDR) of 45 dB upon 914 nm illumination, suggesting an excellent linear dynamic characteristic. Our results indicate that assembling Al$_2$O$_3$/quasi-2D perovskite heterostructure dielectric for hybrid phototransistors is highly valuable in solving the hysteresis issue together with achieving both ultrahigh responsivity and excellent LDR.

## Results

**Device fabrication and characterization.** Figure 1a shows the schematic diagram of our heterogeneous structure phototransistor consisting of Al$_2$O$_3$/2D perovskite heterostructure dielectric and multilayer MoS$_2$ conducting channel. The devices were fabricated by exfoliating multilayer MoS$_2$ flakes (~5 nm thick) onto a p-doped Si substrate covered with ~300 nm SiO$_2$ layer. Cr/Au (15/50 nm) electrodes were patterned on MoS$_2$ flakes using e-beam lithography (EBL) and metal thermal evaporation. After atomic layer deposition (ALD) of 9 nm Al$_2$O$_3$, 2D perovskite (PEA)$_2$(MA)$_{n-1}$Pb$_n$I$_{3n+1}$ was deposited by the one-step spin coating method. Here, PEA$^+$ represents C$_8$H$_9$NH$_3^+$, MA$^+$ refers to CH$_3$NH$_3^+$. Subsequently, 20 nm Al electrode was formed on top of 2D perovskite by thermal evaporation through a shadow mask. The scanning electron microscopy (SEM) image of a typical device is shown in Fig. 1b. Meanwhile, Fig. 1c gives the corresponding Raman spectra of multilayer MoS$_2$ flake for all structures. There is no obvious variation in peak positions, indicating that the present Al$_2$O$_3$/2D perovskite dielectric engineering does not introduce any detectable bond-disorder or lattice damage to MoS$_2$ channel. The absorption spectrum demonstrates that spin-coated 2D perovskite films can contain multiple phases of different $n$ values (Supplementary Fig. 1), which is in accordance with previous studies[19,20]. To examine whether the perovskite phases distribution in 2D perovskite film follows a specific order or in a random configuration, photoluminescence (PL) spectra measurements were carried out with two different configurations (inset of Fig. 1d). Here, the laser beam of 457 nm is either illuminated through the 2D perovskite (i.e., front excitation) or the glass substrate (i.e., back excitation). A dominant spectra at 764 nm ($n \approx \infty$) was observed in both excitations; however, four emission peaks at higher energy ($n = 1$–4) under back-excitation were observed, which implies that the large-$n$ phases should majorly locate near the upper side and the small-$n$ phases locate near the bottom side of the spin-coated 2D perovskite film. This particularly ordered phase distribution within perovskite results in a type-II band alignment along the direction perpendicular to the substrate[21].

As shown in Fig. 1e, upon illumination, a self-driven photo-generated carriers separation process is supported by energy band alignment in the heterostructure dielectric, which can prolong the carriers lifetime ($\tau_{\text{lifetime}}$) and subsequently induces a high photoconductive gain according to the following equation[22]:

$$G = \frac{\tau_{\text{lifetime}}}{\tau_{\text{transit}}}, \tag{1}$$

where $\tau_{\text{transit}}$ refers to transit time of the photo-generated carriers transporting within the conducting channel. To clearly verify the improvement of this heterostructure dielectric for highly sensitive photodetection, the photodetectors based on bare MoS$_2$, MoS$_2$/Al$_2$O$_3$, and MoS$_2$/Al$_2$O$_3$/2D perovskite ($n = 3$) with the same MoS$_2$ flake are fabricated and characterized under the same measurement conditions (e.g., laser power density of $P_{\text{light}} = 1232.2$ µW/cm$^2$, wavelength of $\lambda = 914$ nm, and gate voltage of $V_{\text{gs}} = 0$ V). As given in Supplementary Fig. 2, both bare MoS$_2$ and MoS$_2$/Al$_2$O$_3$ devices exhibit negligible photocurrent ($I_{\text{ph}}$), while MoS$_2$/Al$_2$O$_3$/2D perovskite device presents a remarkable

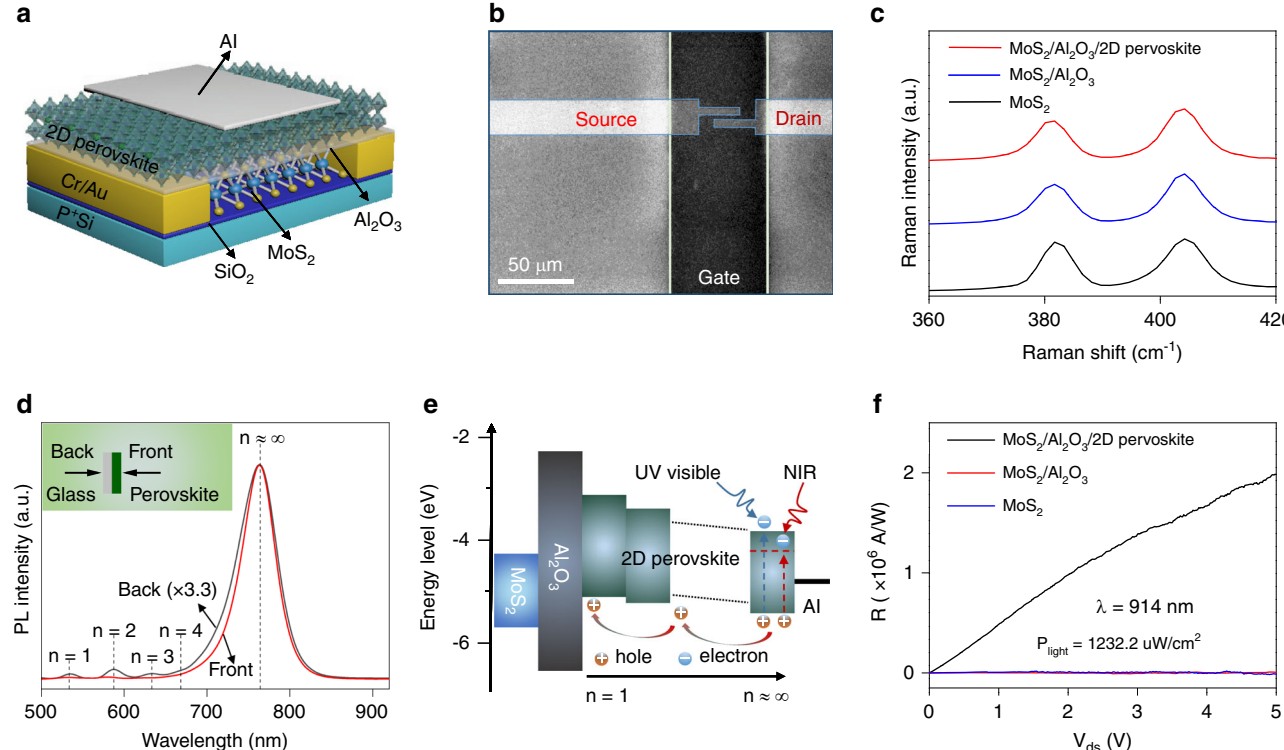

**Fig. 1 MoS$_2$ phototransistor with Al$_2$O$_3$/2D perovskite heterostructure dielectric. a** Schematic diagram of the proposed phototransistor. **b** SEM image of a typical device. **c** Raman spectrum of the bare MoS$_2$, MoS$_2$/Al$_2$O$_3$, and MoS$_2$/Al$_2$O$_3$/2D perovskite structures excited with 532 nm laser. **d** PL spectrum of 2D perovskite (PEA)$_2$(MA)$_2$Pb$_3$I$_{10}$ ($n = 3$) illuminated from different sides of the film. Under front-excitation, the PL spectrum exhibits a dominant emission peak from $n \approx \infty$ phase. In the case of back-excitation, the spectrum shows extra emission corresponding to $n = 1$–4 phases. **e** Schematic of the photo-generated carriers transfer process in spin-coated 2D perovskite films with type-II energy band alignment. **f** Photoresponse of the bare MoS$_2$, MoS$_2$/Al$_2$O$_3$, and MoS$_2$/Al$_2$O$_3$/2D perovskite ($n = 3$) photodetectors measured at $P_{light} = 1232.2$ μW/cm$^2$ and $V_{gs} = 0$ V upon 914 nm illumination.

photoresponse. According to the equation[23]:

$$R = \frac{I_{light} - I_{dark}}{P_{light}S} = \frac{I_{ph}}{P_{light}S}, \qquad (2)$$

where $I_{dark}$ and $I_{light}$ refer to the device output currents with and without illumination, respectively, and $S$ denotes the area of MoS$_2$ channel, the photoresponsivity of MoS$_2$/Al$_2$O$_3$/2D perovskite device is then extracted to be $2 \times 10^6$ A/W. Interestingly, the photoresponse in this NIR region is beyond the 2D perovskite absorption edge, which is probably due to the excitation of photogenerated carriers from valence band to traps states within 2D perovskite bandgap, similar to the extrinsic photoconductors based on normal semiconductors[24]. By employing this Al$_2$O$_3$/2D perovskite heterostructure dielectric, under illumination, the photo-generated holes tend to move towards Al$_2$O$_3$/2D perovskite interface. The spatial separation of photo-generated carriers in 2D perovskite would lead to an increase in the electric field at the MoS$_2$, which could increase the carrier density in the MoS$_2$, thus enhancing the conductivity[25]. Therefore, ultrahigh photoresponsivity in NIR region can be obtained, which differs from the case of photovoltaic photodetectors. After the laser is switched off, the photo-generated carriers would gradually recombine with the opposite charge in 2D perovskite. Because the additional electrons in MoS$_2$ are induced by the accumulation of photo-generated holes at the Al$_2$O$_3$/2D perovskite interface, the electrons in MoS$_2$ would eventually return to the pre-illumination levels[25].

**Hysteresis characteristics of the phototransistor.** Currently, the observed hysteresis issue in perovskite-based devices has attracted

extensive attention because it makes operational stability of perovskite-based devices under a bias stress into a major challenge for those envisioned applications. Figure 2a exhibits the typical transfer characteristic curves of MoS$_2$-based phototransistors with different gate dielectrics. The $V_{gs}$ scanning direction is scanned from negative to positive, and then back to negative voltages. The corresponding leakage current ($I_{gs}$) curves are shown in Supplementary Fig. 3. Apparently, for the device with single Al$_2$O$_3$ dielectric, a clockwise hysteresis loop is observed, which is caused by the trapping of negative charge from the gate-induced conduction channel into immobile localized state located at MoS$_2$/Al$_2$O$_3$ interface. On the other hand, in comparison with 3D perovskites, the carriers hopping barrier of insulating RNH$_3$ bilayer in 2D perovskites offers them opportunity for dielectric application (Supplementary Fig. 4 and Table 1). However, the typical MoS$_2$ transistor with single 2D perovskite dielectric exhibits an anti-clockwise hysteresis loop behavior (Supplementary Fig. 4), which cannot be attributed to the trapping of electrons into localized state. It is well known that the external electric field would induce a migration of ionic vacancies ($V_{PEA}$, $V_{MA}$, $V_{Pb}$, and $V_I$ standing for PEA$^+$, MA$^+$, Pb$^{2+}$, and I$^-$ vacancies) in perovskites[16,26]. For MoS$_2$ transistor with 2D perovskite dielectric, the gate bias-induced movement of ionic species may result in a directional electric field, which subsequently causes additional charge accumulation in MoS$_2$ channel. Therefore, the device threshold voltage ($V_{th}$) shifts towards the negative direction. According to this speculation, by modulating the activation energy for ionic migration in 2D perovskites, the two negative effects of ionic migration in perovskite and charge trapping in Al$_2$O$_3$ may neutralize with each other. In our experiments, negligible hysteresis is obtained in MoS$_2$ transistor

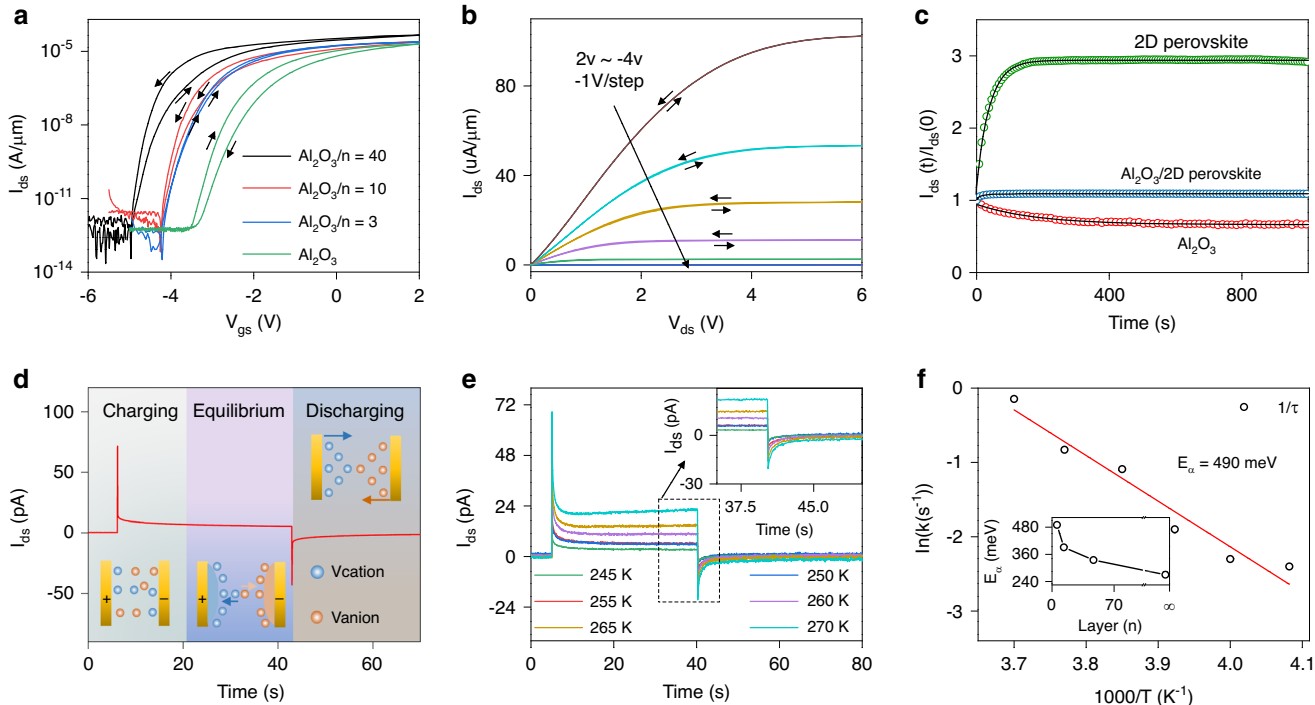

**Fig. 2 Hysteresis characteristics of Al$_2$O$_3$/2D perovskite heterostructure dielectric. a** Transfer characteristic curves of MoS$_2$ phototransistors at $V_{ds}=1$ V with Al$_2$O$_3$ and Al$_2$O$_3$/2D perovskite ($n=3$, 10, and 40) dielectrics. The neglectable hysteresis loop is achieved by using Al$_2$O$_3$/(PEA)$_2$(MA)$_2$Pb$_3$I$_{10}$ ($n=3$) heterostructure dielectric. **b** Output characteristic curves of the MoS$_2$ phototransistor with Al$_2$O$_3$/(PEA)$_2$(MA)$_2$Pb$_3$I$_{10}$ ($n=3$) heterostructure dielectric. **c** Comparisons of the drain-source current ($I_{ds}$) evolution over time under continuous gate bias stress ($V_{gs}=2$ V) for three devices. **d** Schematic illustration of the charging–discharging process for Au/2D perovskite/Au device. **e** Temporal response curves for Au/2D perovskite/Au device measured under dark. **f** Arrhenius plot of the ion decay rate $1/\tau$. The solid line represents the fitting result. Inset: The activation energy of 2D perovskite with different $n$ values.

with Al$_2$O$_3$/(PEA)$_2$(MA)$_2$Pb$_3$I$_{10}$ ($n=3$) heterostructure dielectric by simply modulating the $n$ value of 2D perovskite. In addition, the output curves of MoS$_2$ transistor with Al$_2$O$_3$/(PEA)$_2$ (MA)$_2$Pb$_3$I$_{10}$ heterostructure dielectric also show negligible hysteresis (Fig. 2b), while the device with single Al$_2$O$_3$ or (PEA)$_2$ (MA)$_2$Pb$_3$I$_{10}$ dielectric exhibits a clockwise or anti-clockwise hysteresis loop (Supplementary Fig. 5). To further demonstrate our speculation, we plot drain-source current ($I_{ds}$) evolution over time under continuous gate bias stress ($V_{gs}=2$ V) for three devices in Fig. 2c. Apparently, the $I_{ds}$ decay with Al$_2$O$_3$ dielectric fits well with the stretched exponential function for trapping effect of negative charge[27,28]:

$$I_{ds}(t) = I_0(0) \exp\left[-\left(\frac{t}{\tau_d}\right)^\alpha\right], \quad (3)$$

where $I_{ds}(0)$ denotes the initial $I_{ds}$ value measured at time $t=0$, $\alpha$ denotes the dispersion parameter, and $\tau_d$ denotes the characteristic time constant for charge trapping. By contrast, the $I_{ds}$ value with 2D perovskite dielectric first increases and then gradually saturates. Here, we modify the above-mentioned stretched exponential function to describe the $I_{ds}$ decay caused by the perovskite layer polarization associated to ionic migration[29,30]:

$$I_{ds}(t) = I_0(0)\left\{1 - \exp\left[-\left(\frac{t}{\tau_p}\right)^\beta\right]\right\}, \quad (4)$$

where $\beta$ refers to the dispersion parameter, and $\tau_p$ represents the characteristic time constant for ionic migration. Here, charge trapping and ionic migration are independent phenomena. Accordingly, we can fit $I_{ds}$ decay of Al$_2$O$_3$/(PEA)$_2$(MA)$_2$Pb$_3$I$_{10}$

device with the equation acquired by the sum of Eqs. (3) and (4):

$$I_{ds}(t) = I_0(0)\left[\left\{1 - \exp\left[-\left(\frac{t}{\tau_p}\right)^\beta\right]\right\} + \exp\left[-\left(\frac{t}{\tau_d}\right)^\alpha\right]\right]. \quad (5)$$

The fitting result demonstrates the combined effect of ionic migration and charge trapping in stabilizing perovskite-based devices.

On the other hand, as shown in Fig. 2a, the anticlockwise hysteresis loop of Al$_2$O$_3$/2D perovskite device increases with increased $n$ values, which suggests the enhanced ionic migration. This is probably due to the reduction of PEA organic spacers in 2D perovskites. The studies to date indicate that the primary ionic migration species in OHPs is $V_I$ due to its relatively small activation energy[16]. Here, in order to quantitatively analyze the ionic migration process in 2D perovskite, we utilize a temperature-related transient response measurement method demonstrated by Duan and co-workers[26]. Briefly, as an external bias is applied to a vertically stacked Au/2D perovskite/Au device, both electrons/holes and ionic vacancies in 2D perovskite start to drift (Fig. 2d). The observed instant current spike with the applied bias is induced by the fast drift of electrons/holes along the external electric field. Meanwhile, the ionic vacancies also gradually accumulate at 2D perovskite/Au interfaces with time, leading to an internal electric field opposite to the external electric field. This ion-induced electric field would then reduce the electrons/holes current continually until the equilibrium condition reached. After the applied bias is removed, the ionic vacancies would migrate backwards due to the large ions concentration gradient, resulting in a negative current. Therefore,

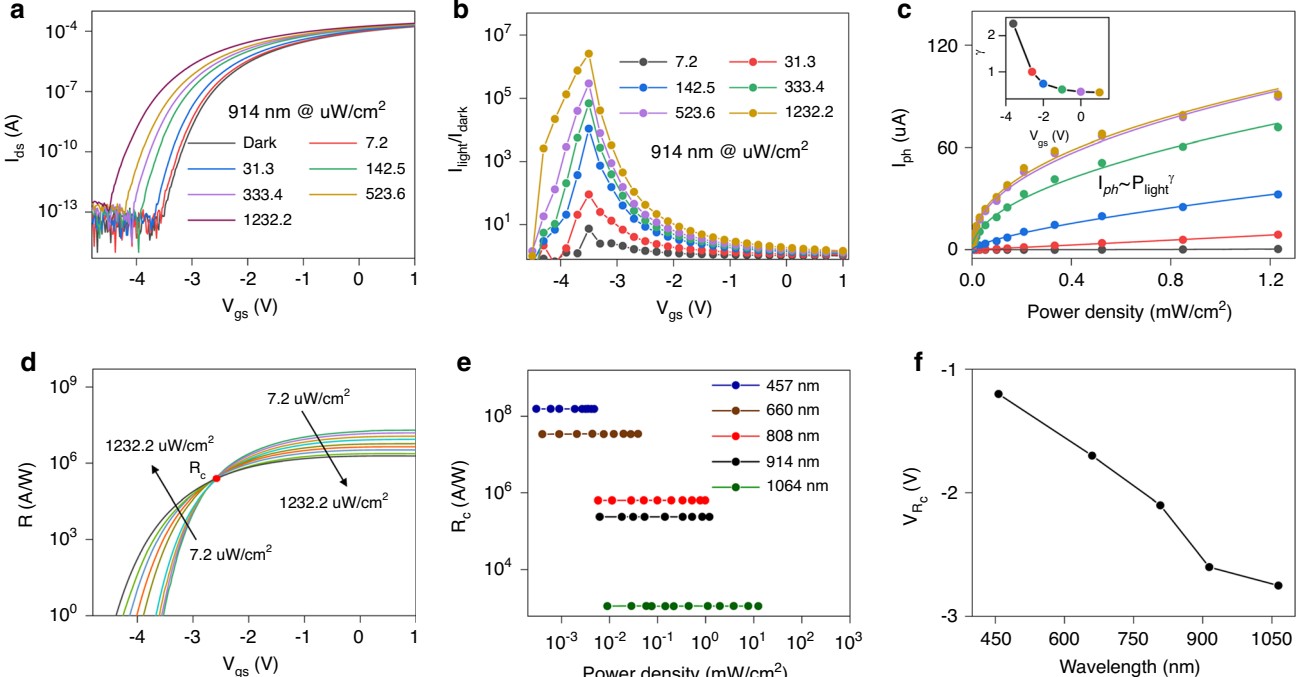

**Fig. 3 Optoelectronic characteristics of the MoS₂ phototransistor. a** Transfer characteristic curves of the device with $Al_2O_3/(PEA)_2(MA)_2Pb_3I_{10}$ ($n = 3$) dielectric measured at $V_{ds} = 1$ V under different laser power density. **b** The corresponding light-to-dark current ratio extracted from **a**. **c** Plots of the photocurrent versus laser power density. Inset: Exponent ($\gamma$) extracted from **c** for each $V_{gs}$. **d** Gate voltage and $P_{light}$ dependent photoresponsivity extracted from **a**. **e** The extracted constant photoresponsivity regardless of laser power density under different illumination levels ($\lambda$: 457, 660, 808, 914 and 1064 nm). **f** Plots of the gate voltage at the turning point versus incident wavelength.

the decay rate of the negative current fully reflects the ionic migration kinetics in 2D perovskite (Fig. 2e). We can thus extract the $E_\alpha$ value (Fig. 2f; Supplementary Fig. 6). The activation energy for 2D perovskites are calculated to be 270, 335, 391, and 490 meV at $n = \infty$, 40, 10, and 3, respectively, suggesting that the ionic migration in our $Al_2O_3$/2D perovskite heterostructure dielectric can be optimized by modulating long-chain PEA cations in 2D perovskites. We further employ first principle calculation to simulate energy barrier for $V_I$ diffusion in 2D perovskite crystal (Supplementary Fig. 1). The calculated energy barrier for interlayer diffusion of $V_I$ is 2.7 eV, much larger than that of 0.7 eV for inner-layer diffusion, indicating the distinct interlayer migration restraining effect of the organic ammonium layers.

**Optoelectronic characteristics and mechanism analysis.** Based on the $Al_2O_3$/2D perovskite heterostructure dielectric with desirable negligible hysteresis, photodetection performances of the device were characterized in detail. Figure 3a depicts the transfer characteristic curves of the device under 914 nm illumination at $V_{ds} = 1$ V. The corresponding $I_{gs}$ curves are shown in Supplementary Fig. 8. Apparently, the threshold voltage of MoS₂ phototransistor shifts towards a more negative direction as the laser incident power increases. In view of the $n$ type conductive characteristic of MoS₂, this result demonstrates the trapping of photo-generated holes at $Al_2O_3$/2D perovskite interface, which can electrostatically modulate the MoS₂ channel and lead to the electron concentration increment there. Figure 3b exhibits the light-to-dark output current ratio ($I_{light}/I_{dark}$) extracted from transfer curves under different incident power, which is an important parameter for obtrusive noise evaluation. Note that the $I_{light}/I_{dark}$ value dramatically increases with the increased incident power, reaching an ultrahigh value of $2.6 \times 10^6$ at $V_{gs} = -3.5$ V even when the $P_{light}$ as low as ~1 mW/cm². In dark, the device

operates in off-state and the current is very low. With light illumination, the device turns on due to the negative shift of threshold voltage and the current increases significantly. The extracted $I_{light}/I_{dark}$ value is much higher than previous reported perovskite-based phototransistor under the same laser power density, indicating a profound photogating effect in our device[31].

We find that both incident power and gate bias influence the device photoresponsivity. As shown in Fig. 3c, the photocurrent increases with the increased incident power, following an equation with the form of $I_{ph} \sim P_{light}^\gamma$, where $\gamma$ is a constant. The nonunity exponent of $0 < \gamma < 1$ is often observed in photogating devices[32,33], as a result of the complex process of carrier generation, trapping, and recombination within semiconductors. The photocurrent tends to saturate as the increase of the laser power, which is partly due to the gradually filled trap states. The smaller $\gamma$ value represents the more prominent photogating effect, while $\gamma = 1$ represents the pure photoconductive effect. However, a high $\gamma$ value of 2.3 at $V_{gs} = -3.5$ V is observed in our device. On the other hand, using the Eq. (2), the extracted $R$ values are shown in Fig. 3d. Here, we can observe a clear turning point regarding $R \sim P_{light}$ dependence at $V_{gs} = -2.7$ V. Specifically, when $V_{gs}$ value is larger than $-2.7$ V, the $R$ value gets decreased as incident power increases. On the contrary, when $V_{gs}$ value is smaller than $-2.7$ V, the $R$ value is observed to increase with increased incident power. This phenomenon is possibly caused by the external biasing field-dependent photo-generated carriers recombination in 2D perovskite. We will discuss it in the following section. Upon 914 nm illumination, the highest $R$ value at $P_{light} = 7.2$ μW/cm² is extracted to be $2.0 \times 10^7$ A/W. This NIR photoresponse is expected to be further enhanced by means of using 2D perovskites with narrower band gap as photoactive dielectrics, integration with metallic plasmonic nanostructure resonated at NIR region, as well as doping 2D perovskites to introduce appropriate trap states within the band gap of 2D perovskites. Due to the much

higher absorption coefficient of 2D perovskite at 457 nm, the highest $R$ value is extracted to be $5.5 \times 10^8$ A/W. In this regard, the photogain of this device can be estimated with the equation[34]:

$$G = \frac{I_{ph}/q}{P_{light}/h\nu} = \frac{Rhc}{q\lambda}, \tag{6}$$

where $h$ represents Planck's constant, $\nu$ represents the incident light frequency, and $c$ stands for the light speed. An ultrahigh photogain of $1.4 \times 10^9$ is then obtained. For certain applications, such as illumination meters and image sensors, photodetectors are expected to have a constant responsivity over a wide range of incident light intensity. As a figure-of-merit for photodetectors, LDR is widely used to characterize the incident light intensity range in which the photodetectors have a constant responsivity, which can be expressed as[35]:

$$LDR = 20\log\frac{p_{sat}}{p_{low}}, \tag{7}$$

where $P_{low}$ ($P_{sat}$) denotes the incident light intensity when $P_{light}$ weaker (stronger) than which the photocurrent starts to deviate from linearity. Upon 914 nm illumination and at $V_{gs} = -2.7$ V, the photoresponsivity remains almost a constant of $2.4 \times 10^5$ A/W within the $P_{light}$ value ranges from 7.2 to 1232.2 $\mu$W/cm$^2$, corresponding to an excellent LDR value of 45 dB. By fitting the data linearly, we found that the R-squared (coefficient of determination) of the linear fitting for the device is 0.998, which is very close to 1, indicating the perfect linearity of the data. As $P_{light}$ value increases continuously, the $I_{ph}$ value starts to deviate from linear relationship with $P_{light}$, reaching the saturation power. In addition, the same measurements with illumination of 457, 660, 808, and 1064 nm were also performed, which exhibit similar phenomenon (Supplementary Fig. 9). The extracted $R_c$ values are shown in Fig. 3e. These constant $R$ values are several orders of magnitude higher than previous reported photodetectors with linear dynamic characteristic[35,36]. Meanwhile, the corresponding LDR values are extracted to be 24, 38, 46, and 82 dB with illumination of 457, 660, 808, and 1064 nm, respectively. Figure 3f presents the gate voltage at the turning point ($V_{Rc}$) under different incident light. Interestingly, it is observed that the $V_{Rc}$ is dependent on incident wavelength, in which the turning point voltage typically shifts towards more negative direction under long wavelength incident light.

The constant photoresponsivity in MoS$_2$ phototransistors with Al$_2$O$_3$/2D perovskite heterostructure dielectric leads to a speculation that the external biasing field-dependent photo-generated carriers recombination in 2D perovskite induces this linear dynamic characteristic. As shown in Fig. 4a, a gate bias lower than $V_{Rc}$ but higher than the voltage where the device is in off state would induce a slight band bending in 2D perovskite. In this case, the electric field experienced by 2D perovskite is relatively small. As illuminated with a weak light, the photon penetration depth is very shallow due to the large light absorption coefficient of perovskite. In view of the high surface charge trap density in OHPs[37–39], therefore, the photo-generated carriers can be easily quenched (Fig. 4b). In comparison, under a strong light excitation, photon penetration depth increases, and part of photo-generated holes can drift to Al$_2$O$_3$/2D perovskite interface instead of recombination (Fig. 4c). Accordingly, the photo responsively increases with increased laser power density. On the other hand, a gate bias higher than $V_{Rc}$ would result in a steep band bending in 2D perovskite (Fig. 4d). The applied gate biasing field can drive carriers spatial separation quickly, making the photo-generated carriers less susceptible to the surface recombination. A weak light excitation thus can induce a remarkable

photoresponse (Fig. 4e). However, as the $P_{light}$ value increases, the increased hole density at Al$_2$O$_3$/2D perovskite interface can induce an inverse built-in electric field, which would reduce carriers collection efficiency. As a result, the photo responsively decreases with increased laser power density. Based on this mechanism, a constant responsively can be achieved by employing an appropriate gate bias. In addition, the observed wavelength dependence of $V_{Rc}$ value is probably due to the difference in photon penetration depth. With short wavelength photoexcitation, photon penetration depth is much shallower due to the much higher absorption coefficient. Therefore, a stronger gate biasing field is required for photo-generated carrier separation, corresponding to a more positive $V_{Rc}$ value.

Figure 5a exhibits the switching performance of the photo-transistor upon 914 nm illumination measured at $P_{light} = 523.6$ $\mu$W/cm$^2$ and $V_{gs} = -3.5$ V. There is negligible variation in photocurrent, indicating the stable and reversible response of our device. The rise (decay) time ($\tau_{rise}$ ($\tau_{decay}$)) here is defined as the period for photocurrent to rise (decay) from 10 to 90% (90 to 10%) of the final value. Therefore, according to a high resolution scan in one cycle of switch curve, the $\tau_{rise}$ and $\tau_{decay}$ are estimated to be 27 and 29 ms, respectively (Fig. 5b). However, the response speed of the device gets slower as the gate bias increased (Supplementary Fig. 10), probably due to the suppressed carriers recombination with the stronger gate biasing field. Additionally, the stability of the phototransistor with Al$_2$O$_3$/2D perovskite dielectric was also investigated. Due to the hydrophobicity of organic spacer, the devices exhibit the excellent stability. The photoresponsivity retain 98 and 89% of its original values after exposing in nitrogen and air environment for more than 160 h, respectively (Supplementary Fig. 11). Besides responsivity, another key performance parameter for photodetectors is specific detectivity ($D^*$), which describes the capability of the device in detecting a weak signal, can be calculated using the equation of[34,40]:

$$D^* = \frac{(SB)^{1/2}}{NEP}, \tag{8}$$

$$NEP = \frac{\overline{i_n^2}^{1/2}}{R}, \tag{9}$$

where $B$ refers to the bandwidth, $\overline{i_n^2}^{1/2}$ refers to the root mean square value of the spectral noise density, and NEP refers to the noise equivalent power. Figure 4c presents the full set of $\overline{i_n^2}^{1/2}$ values measured under different gate bias to determine the $D^*$ value. A considerable decrease of $\overline{i_n^2}^{1/2}$ of orders of magnitude for more negative gate bias is observed. The $\overline{i_n^2}^{1/2}$–$V_{gs}$ curve shows a clear $1/f$ component at $V_{gs} = -2.7$ V, while the measuring instrument reaches its noise floor at $V_{gs} = -3.5$ V. Despite the highest responsivity at $V_{gs} = 1$ V, the $D^*$ value is relatively low due to the large noise value (Fig. 5d). The constant $D^*$ values are calculated to be $1.3 \times 10^{13}$, $9 \times 10^{12}$, $7.2 \times 10^{12}$, $5.5 \times 10^{12}$, and $2.3 \times 10^{11}$ Jones at a frequency of 1 Hz upon 457, 660, 808, 914 and 1064 nm illumination, respectively. Such ultrahigh detectivity thus providing the quantitative evidence that the phototransistor configuration presented here is extremely suitable for the detection of small optical signals.

## Discussion

In summary, we have demonstrated that the MoS$_2$ photo-transistors with Al$_2$O$_3$/2D perovskite heterostructure dielectric allow broadband photoresponse of 457–1064 nm, ultrahigh photogain of $1.4 \times 10^9$, excellent linear dynamic characteristic and high reliability operation. By simply modulating the activation

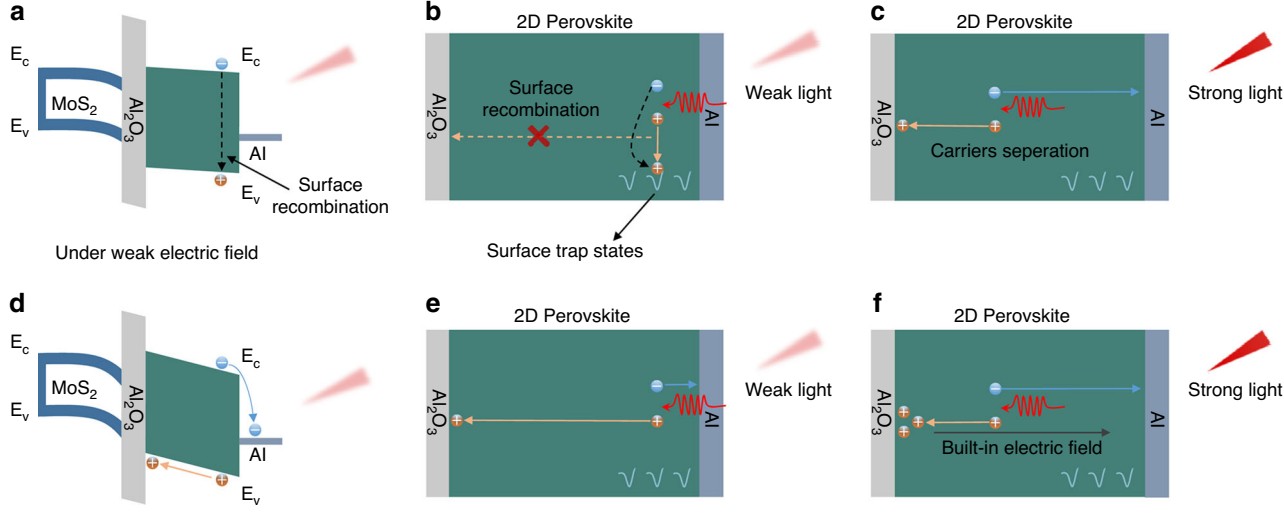

**Fig. 4 Schematic illustrations of the constant photoresponsivity. a** Energy band diagram of the device under a gate bias lower than $V_{Rc}$ but higher than the voltage where the device is in off state. The slight band bending in 2D perovskite cannot promote the photo-generated carriers separation effectively. **b** As illuminated with a weak light, trap-assisted photo-generated carriers recombination would occur at 2D perovskite/Al electrode interface. **c** A stronger light excitation would induce a deeper photo penetration depth. Subsequently, part of photo-generated carriers can diffuse to $Al_2O_3$/2D perovskite interface before recombination. Therefore, the photo responsively increases with increased laser power density. **d** Energy band diagram of the device under a gate bias higher than $V_{Rc}$. The steep band bending in 2D perovskite is beneficial for carriers separation. **e** Even illuminated with a weak light, the applied strong electric field can drive carriers separation effectively. **f** Under a strong light excitation, the accumulated holes at perovskite/Al electrode interface would induce an inverse built-in electric field, and thus reducing carriers collection efficiency. Accordingly, the photo responsively decreases with increased laser power density.

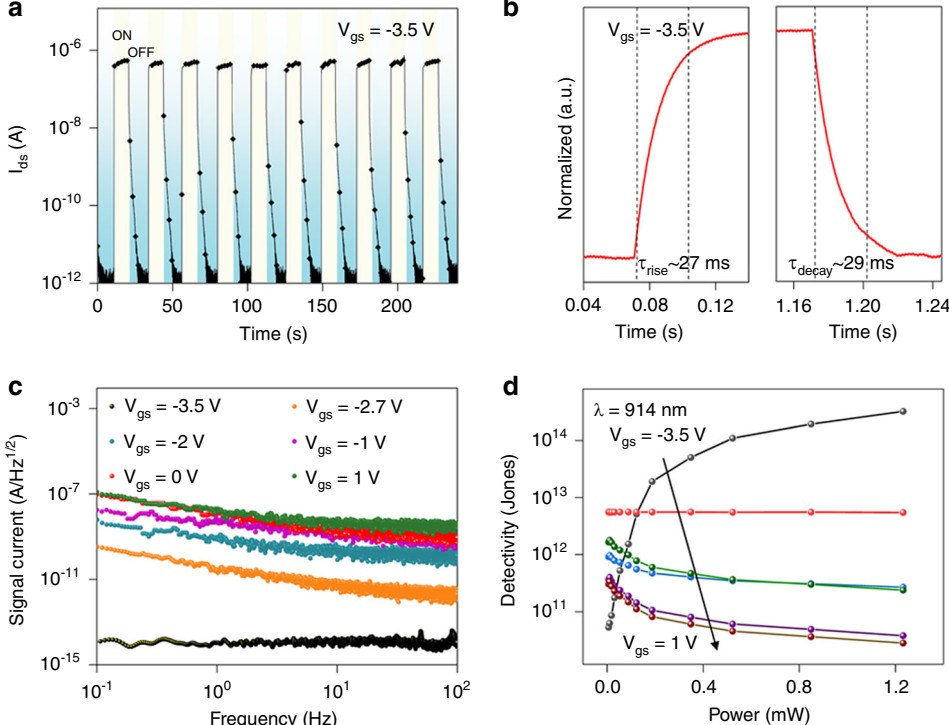

**Fig. 5 Transient properties and detectivity analysis of the MoS₂ phototransistor. a** Switching characteristic of the phototransistor measured at $\lambda = 914$ nm, $P_{light} = 523.6 \, \mu W/cm^2$, $V_{ds} = 1$ V and $V_{gs} = -3.5$ V. **b** The rise and decay time of the phototransistor recorded by oscilloscope. **c** Noise analysis of the phototransistor. The curves exhibit clear $1/f$ component which is not visible as reaching the system noise floor at $V_{gs} = -3.5$ V. **d** Specific detectivity of the phototransistor as a function of laser power density and gate voltage.

energy for ionic migration in 2D perovskites, the two negative effects of ionic migration in perovskite and charge trapping in $Al_2O_3$ can neutralize with each other, resulting in a neglectable hysteresis. It should be noted that perovskite-based photodetectors have been extensively explored in recent years, with most efforts focusing on utilizing perovskite as conducting channel. In contrast to these earlier photodetectors, the $Al_2O_3$/2D perovskite heterostructure dielectric greatly enhances the photoconductive gain of obtained devices. The designed $Al_2O_3$/2D perovskite heterostructure dielectric has great potential for realizing new-generation high-performance photodetectors compatible with traditional microelectronics.

## Methods

**Device fabrication**. Few-layer $MoS_2$ flakes were mechanically exfoliated from $MoS_2$ bulk crystals and transferred to the precleaned p-type silicon substrate covered with 300 nm thick $SiO_2$ layer. Then the substrates were spin-coated with polymethyl methacrylate, and the EBL (JEOL 6510 with NPGS) was used to define the source/drain patterns. The Cr/Au (15/50 nm) electrodes were deposited by metal thermal evaporation and lift-off processes. The channel length is 3 μm, and the channel width is 3–10 μm. Subsequently, a 9-nm thick $Al_2O_3$ was deposited using ALD (precursor: water and trimethylaluminum; deposition temperature = 95 °C). The precursor solutions of different-dimensional perovskite $(PEA)_2(MA)_{n-1}Pb_nI_{3n+1}$ were prepared by dissolving specific stoichiometric quantities of MAI, PEAI and $PbI_2$ in anhydrous N,N-dimethylformamide solvent. The resulting solution was continuously stirred at 50 °C for 12 h inside a nitrogen-filled glovebox and then filtered through Poly tetra fluoroethylene syringe filter. To fabricate the $Al_2O_3$/2D perovskite heterostructure dielectric, the 2D perovskite precursor solution was spin coated on $Al_2O_3$ layer at 4000 rpm for 1 min and annealed at 100 °C for 60 min. Finally, 20 nm Al electrode was deposited on top of 2D perovskite by metal thermal evaporation with a shadow mask.

**Material and device characterizations**. The cross-sectional morphology of the 2D perovskite film was characterized by SEM using JEOL 6510. The absorption spectrum was recorded with Shimadzu UV-2550 spectrometer. PL and Raman spectra were characterized on Renishaw Raman spectrometer platform. Electrical and optoelectrical measurements were performed by using Agilent B1500A semiconductor parameter analyzer. The light sources were lasers with wavelengths of 457, 660, 808, 914, and 1064 nm, respectively.

## Data availability

The data that support the findings of this study are available from the corresponding author upon reasonable request.

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

## Acknowledgements

This work is financially supported by the National Key R&D Program of China (No. 2018YFA0703700), National Natural Science Foundation of China (Grant nos. 61925403, 61851403, 61811540408, 51872084, and 61704051), the Strategic Priority Research Program of Chinese Academy of Sciences (Grant no. XDB30000000), as well as the Natural Science Foundation of Hunan Province (Nos. 2020JJ1002).

## Author contributions

X.Z. and L.L. conceived the concept and experiments. X.Z., Y.C., and Y.L. prepared the paper. J.J. fabricated and characterized the devices. W.X. and Q.T. assisted the experiments on device fabrication and measurements. Theoretical calculation is finished by Y.W.L. All authors examined and commented on the paper.

## Competing interests

The authors declare no competing interests.
