## [Peer Review File · Nature Communications]

Reviewers' comments:

Reviewer #1 (Remarks to the Author):

Lei Liao et al. report high optoelectronic performance MoS₂ transistors using a hybrid Al₂O₃/2D perovskite gate dielectric approach. Specifically, they attempt to show how net hysteresis close to zero can be achieved by choosing the perovskite layer such that the anti-clockwise hysteresis due to ionic movement in the perovskite cancels out the clockwise hysteresis due to negative charge trapping in the Al₂O₃ dielectric. They also show stable dark current with the hybrid dielectric, high photoresponse over visible-near IR range, good LDR and detectivity, and switching speed of the order of 20-30 ms.

Although the photoresponsivity values, LDR and broad spectral response metrics are impressive, not as much the switching speed, there are substantial concerns regarding the motivation, proposed mechanism and supporting data. The claims are novel but not scientifically convincing as outlined in the questions below.

(i) Accumulation of photogenerated holes at the Al₂O₃/perovskite interface leads to an electron current in the MoS₂ through an electrostatic gating effect. If this is the correct understanding, then what is the motivation for choosing MoS₂, why not Si or any other conventional semiconductor with much higher electron mobility than MoS₂?

(ii) The photoresponse at 914 nm is beyond the absorption edge of the perovskite- the authors attribute this to excitation from valence band to trap states (lines 136-140).

(a) Isn't this inconsistent with Fig. 1e where electrons are shown in the conduction band? Where are the trap states in this picture?

(b) If photogenerated electrons are captured by traps from the valence band leading to photogenerated holes in the valence band- how do these electrons move towards the gate electrode under external gate bias as shown in Fig. 1e?

(iii) What are the mechanisms behind recombination of photogenerated holes in 2D perovskite and electrostatically induced "supplemental" electrons in MoS₂ - in their respective layers?

(iv) The authors claim that the hysteresis due to Al₂O₃ cancels that due to the 2D perovskite. However, voltage shifts due to dielectric charge need not be simply additive/subtractive, since the shift does not depend on charge density but the moment of charge density. Hence the contribution to hysteresis due to positive ionic charges in the perovskite will be less once it is placed on top of Al₂O₃. Comparing the hysteresis in Fig. S5 due to perovskite (n=3) alone and due to Al₂O₃ alone (Fig. 2a), it seems unlikely that these will cancel out in the composite stack. Please clarify.

(v) Comparing the dark transfer curves (I_{ds}-V_{gs}) in Fig. 2a for Al₂O₃/n=3, Al₂O₃ only and only n=3 in Fig. S5 one comes up with the following inconsistencies/questions:

(i) The threshold voltage of n=3 is nearly 0 V, that of only Al₂O₃ is ~-2 V and that of Al₂O₃/n=3 is nearly -3 V. This seems to indicate net fixed positive charge in Al₂O₃. How did it increase with n=3 that seems to have no net fixed charge ($V_T \sim V_{flatband}$)?

(ii) n=3 has poor sub-threshold slope (visually) and on-current. How did these improve with the addition of a 9 nm dielectric in series? Please specify quantitative values of carrier mobility, subthreshold slope, V_T , hysteresis, I_{on} and I_{off} for all three key transistors- Al₂O₃/n=3, Al₂O₃ only and only n=3, maybe in a table in the supplementary information.

(vi) What could be the reason for a high gamma value of 2.3?

(vii) The energy band diagram in Fig. 4a under $V_g < V_{Rc}$ does not seem to be correct. As per Fig. 3a the transistor is below threshold (close to being off) at large negative V_g . In this case the band bending should be in the opposite direction than what is shown, and MoS₂ is depleted (not

accumulated as shown). The electric field in the perovskite will be in the opposite direction leading to electron accumulation at Al₂O₃/perovskite interface.

(viii) The explanation for VRC crossover is not clear- if it is due to increasing electric field in the perovskite, what determines the critical electrical field- corresponding to VRC, in terms of perovskite material parameters, where the light intensity dependence of R flips over? Further, if a built-in electric field due to hole accumulation opposes further hole drift to the interface (Fig. 4f), shouldn't the R value saturate for increasing light intensity?

(ix) Can the authors show stability of optical photoresponse over a period of time?

(x) There are a large number of writing errors, the manuscript needs thorough proof reading. A few examples:

(a) line 206 - "stack" should be "stacked"

(b) line 168- "charges" should "charge"

(c) line 92 - "consisted" should be "consisting"

Reviewer #2 (Remarks to the Author):

In this work, the authors present a simple yet universal Al₂O₃/2D perovskite heterostructure dielectric for high-performance phototransistors. The type-II energy band alignment in 2D perovskite facilitates the effective spatial separation of photoexcited carriers, thus inducing the outstanding photoresponse properties. By adding a high-k Al₂O₃ layer between the 2D perovskite layer and the semiconductor layer, the two effects of charge trapping and remnant polarization under gate bias are found to be neutralized with each other, resulting in low-voltage transistors of negligible hysteresis and improved reliability. The devices also exhibit excellent linear dynamic characteristics.

Overall, I think this paper is sufficiently innovative and the experiments are nicely executed. I think the work will be of interest to diverse researchers working on perovskite materials and 2D-material phototransistors and I would recommend its publication in Nature Communications after the following questions are clarified.

(1) Although there has been significant progress on the development of perovskite materials in recent years, the poor stability is still a grand challenge yet to be tackled. How's the stability of the devices reported in this paper and what are the possible approaches to improve the lifetime of perovskite-based devices? Some discussions would be helpful.

(2) The device is capable of broadband photodetection ranging from 457 nm to 1064 nm. However, the absorption edge of the upper perovskite layer is located at around 800 nm. Is the MoS₂ layer responsible for the near-infrared response?

(3) Page 12, Line 5, the authors estimated the $I_{\text{light}}/I_{\text{dark}}$ ratio to be 2.6×10^6 at $V_{\text{gs}} = -3.5$ V where the incident power density (P_{light}) is ~ 1 mW/cm². The electrical measurements were carried out by varying the intensity from ~ 0.01 - ~ 1 mW/cm². In this case how can the author say that P_{light} is as low as 1 mW/cm²?

(4) What are the channel length and width of the phototransistors?

(5) First of all, Fig. 2b has not been mentioned in the manuscript. Furthermore, I assumed the arrows in Fig. 2b is for double sweep to show hysteresis (or lack thereof) in the I_{ds} - V_{ds} characteristics of the phototransistors. Usually people present hysteresis on transfer I_{ds} - V_{gs} characteristics of the devices (e.g. Fig. 2a of the paper). It is not clear to me why it is necessary to also present it for the I_{ds} - V_{ds} characteristics.

Reviewer #3 (Remarks to the Author):

Jiang et al. have demonstrated the fabrication of a high performance MoS₂ phototransistor employing a novel Al₂O₃/2D perovskite hybrid dielectric configuration. The results highlight the achievement of both ultrahigh responsivity and excellent linear dynamic range in one device. In addition, this work on Al₂O₃/2D perovskite dielectric to achieve greatly improved operational stability is very crucial for the technology to be applied in practical fields. I appreciate the experimental work which is systematically performed and discussed. Considering that the overall quality and significance of this work, I would like to recommend this work for publication after the authors address the following questions:

1. In the Al₂O₃/2D perovskite hybrid dielectric, the thickness of Al₂O₃ layer is 9 nm. In my understanding, the Al₂O₃ layer is employed as a high-k dielectric, and the thinner thickness could help to enhance the gate capacitance. Does the Al₂O₃ thickness influence the performance of the device?
2. In Figure 3d, the photoresponsivity remains almost constant within the light power density ranging from 7.2 to 1232.2 $\mu\text{W}/\text{cm}^2$. What is the R-squared (coefficient of determination) of the linear fitting for the device?
3. Why can the devices detect near-infrared light? The perovskites are usually employed for visible light detection, which is in accordance with the absorption spectrum shown in Figure S1.
4. Line 8, Page 3. the sentence "...is also become more eminent." should be corrected; Line 7, Page 10. "...and S denotes the active area of photodetector, ..." Compared to the illumination area, which is larger in the experiment setup? The authors should specify this value in the manuscript.
5. Line 16-18, Page 12. "In general, the γ value is in the range of $0 < \gamma < 1$ for photogating devices, ...photoconductive effect³¹." The author should elaborate more on this point, as this value depends also on the surface defects and quality of film.
6. In Figure 2a and Figure 5a, what is the source-drain voltage used in the measurements?

Response to Reviewer # 1:

We thank the referee for careful reading the manuscript and providing a number of precious comments. And we have addressed the comments carefully with listed response below. Meanwhile we have revised our manuscript accordingly.

Lei Liao et al. report high optoelectronic performance MoS₂ transistors using a hybrid Al₂O₃/2D perovskite gate dielectric approach. Specifically, they attempt to show how net hysteresis close to zero can be achieved by choosing the perovskite layer such that the anti-clockwise hysteresis due to ionic movement in the perovskite cancels out the clockwise hysteresis due to negative charge trapping in the Al₂O₃ dielectric. They also show stable dark current with the hybrid dielectric, high photoresponse over visible-near IR range, good LDR and detectivity, and switching speed of the order of 20-30 ms.

Although the photoresponsivity values, LDR and broad spectral response metrics are impressive, not as much the switching speed, there are substantial concerns regarding the motivation, proposed mechanism and supporting data. The claims are novel but not scientifically convincing as outlined in the questions below.

1. Accumulation of photogenerated holes at the Al₂O₃/perovskite interface leads to an electron current in the MoS₂ through an electrostatic gating effect. If this is the correct understanding, then what is the motivation for choosing MoS₂, why not Si or any other conventional semiconductor with much higher electron mobility than MoS₂?

We thank for the referee's detailed comments on these viewpoints. In the device architecture, a conducting channel with high electron mobility would lead to the improved photoresponsivity. But as the scale and diversity of application areas grow, the needs for a photodetection platform with higher integration and lower power consumption are becoming more eminent. In comparison with conventional semiconductors, the ability to control the MoS₂ thickness at the atomic level translates into improved gate control over the channel barrier and into reduced short-channel effects, which are beneficial for ultrascaled devices with low power consumption¹.

2. The photoresponse at 914 nm is beyond the absorption edge of the perovskite the authors attribute this to excitation from valence band to trap states (lines 136-140).

(a) Isn't this inconsistent with Fig. 1e where electrons are shown in the conduction band? Where are the trap states in this picture?

(b) If photogenerated electrons are captured by traps from the valence band leading to photogenerated holes in the valence band- how do these electrons move towards the gate electrode under external gate bias as shown in Fig. 1e?

(a) We thank for the reminder and sorry for the mistake. We have corrected the energy band diagram in the manuscript.

Figure 1. Schematic of the photo-generated carriers transfer process in spin-coated 2D perovskite films with type-II energy band alignment.

(b) In this case, the photo-generated holes tend to move towards Al₂O₃/2D perovskite interface. The spatial separation of photo-generated carriers in 2D perovskite would lead to an increase in the electric field at the MoS₂, which could increase the carrier density in the MoS₂, thus enhancing the conductivity². We have corrected the relevant discussion in the manuscript.

3. What are the mechanisms behind recombination of photogenerated holes in 2D perovskite and electrostatically induced “supplemental” electrons in MoS₂ – in their respective layers?

We are sorry for the confusion. The carrier recombination is probably associated with trap-assisted recombination through the traps or defect levels in Al₂O₃ or perovskite layer³. We have added the relevant discussion in the manuscript.

4. The authors claim that the hysteresis due to Al₂O₃ cancels that due to the 2D perovskite. However, voltage shifts due to dielectric charge need not be simply additive/subtractive, since the shift does not depend on charge density but the moment of charge density. Hence the contribution to hysteresis due to positive ionic charge in the perovskite will be less once it is placed on top of Al₂O₃. Comparing the hysteresis in Fig. S5 due to perovskite (n=3) alone and due to Al₂O₃ alone (Fig. 2a), it seems unlikely that these will cancel out in the composite stack. Please clarify.

We thank for the referee’s detailed comments and sorry for the confusion. In our experiments, a clockwise hysteresis loop is observed for the device using Al₂O₃ gate dielectric, which is caused by the trapping of carriers from the gate bias-induced conduction channel into less mobile localized states. Meanwhile, the fabricated phototransistor using 2D perovskite gate dielectric presents hysteresis loop opposite to that of the device with Al₂O₃ gate dielectric layer. The physics mechanism is

considered to be that the electric field from ionic polarization in 2D perovskite induces additional carriers into the channel. Here, the hysteresis due to perovskite ($n = 3$) alone in the Fig. S5 is associated with the applied gate bias (the Fig. S5 has been merged into Supplementary Fig. 4b). In order to avoid the breakdown of the perovskite ($n = 3$) alone, the applied gate bias is limited. Accordingly, it is hard to compare the hysteresis due to perovskite ($n = 3$) alone and due to Al_2O_3 alone. By adding a thin Al_2O_3 layer between the 2D perovskite and the semiconductor layer, the two effects of charge trapping and ionic polarization under gate bias could be neutralized with each other, resulting in negligible hysteresis in MoS_2 phototransistor.

5. Comparing the dark transfer curves (I_{ds} - V_{gs}) in Fig. 2a for $\text{Al}_2\text{O}_3/n=3$, Al_2O_3 only and only $n=3$ in Fig. S5 one comes up with the following inconsistencies/questions:

(i) **The threshold voltage of $n=3$ is nearly 0 V, that of only Al_2O_3 is ~ -2 V and that of $\text{Al}_2\text{O}_3/n=3$ is nearly -3 V. This seems to indicate net fixed positive charge in Al_2O_3 . How did it increase with $n=3$ that seems to have no net fixed charge ($V_T \sim V_{flatband}$)?**

(ii) **$n=3$ has poor sub-threshold slope (visually) and on-current. How did these improve with the addition of a 9 nm dielectric in series? Please specify quantitative values of carrier mobility, subthreshold slope, V_T , hysteresis, I_{on} and I_{off} for all three key transistors- $\text{Al}_2\text{O}_3/n=3$, Al_2O_3 only and only $n=3$, maybe in a table in the supplementary information.**

(i) We thank for the referee's detailed comments and sorry for the confusion. Here, in order to avoid the error caused by different MoS_2 flakes, the phototransistors are fabricated with the same MoS_2 flake (Figure 2a). The device with Al_2O_3 dielectric exhibits a negative threshold voltage (Figure 2b). Meanwhile, the V_{th} value with Al_2O_3 dielectric is similar to that of $\text{Al}_2\text{O}_3/2\text{D}$ perovskite dielectric. This result can be attributed to the giant dielectric constant phenomenon in 2D perovskite at low frequency (Figure 2c), which is probably caused by intrinsically polarizability of perovskite⁴. In typical series capacitance geometry, the total capacitance is given by:

$$C_{\text{Al}_2\text{O}_3/2\text{D perovskite}} = 1 / (1 / C_{\text{Al}_2\text{O}_3} + 1 / C_{2\text{D perovskite}}) \approx C_{\text{Al}_2\text{O}_3}$$

where $C_{\text{Al}_2\text{O}_3/2\text{D perovskite}}$ is the $\text{Al}_2\text{O}_3/2\text{D}$ perovskite capacitance, $C_{\text{Al}_2\text{O}_3}$ is the Al_2O_3 capacitance, and $C_{2\text{D perovskite}}$ is the 2D perovskite capacitance. For both Al_2O_3 and $\text{Al}_2\text{O}_3/2\text{D}$ perovskite dielectrics, the threshold voltage caused by fixed positive charge in Al_2O_3 (V_{th}') is approximately equal:

$$V_{th}' = \frac{1}{C_{\text{Al}_2\text{O}_3}} \int_0^{t_{\text{Al}_2\text{O}_3}} \frac{x}{t_{\text{Al}_2\text{O}_3}} \rho(x) dx$$

where $t_{\text{Al}_2\text{O}_3}$ is the Al_2O_3 thickness, and $\rho(x)$ is the fixed positive charge density in Al_2O_3 . In addition, the V_{th} value for 2D perovskite dielectric is nearly 0 V, which is probably due to p -type doping effect in the MoS_2 channel and effective electrostatic control.

Figure 2. The fabrication procedure and characterization of MoS₂ phototransistors with three types of dielectrics. **a** Schematic diagrams illustrate the preparation process of the devices. **b** Transfer characteristic curves of MoS₂ phototransistors with Al₂O₃, (PEA)₂(MA)₂Pb₃I₁₀, and Al₂O₃/(PEA)₂(MA)₂Pb₃I₁₀ dielectrics, respectively. **c** Dielectric properties of 2D perovskites.

(ii) We thank for the referee's valuable suggestion. The key performance parameters of the devices are summarized in Table S1. Here, in comparison with Al₂O₃ and Al₂O₃/2D perovskite dielectrics, the device with 2D perovskite dielectric exhibits a much lower electrical performance. It is well known that the interface quality plays a crucial role in the carriers transport of 2D semiconductor devices¹. This result indicates the severe carrier scattering involving surface roughness and Columbic impurity scattering at MoS₂/2D perovskite interface⁵.

Dielectric structure	μ_F/μ_B (cm ² /V·s)	SS_F/SS_B (mV/dec)	V_{thF}/V_{thB} (V)	Hysteresis (V)	I_{on} (A/ μ m)	I_{on}/I_{off}
Al ₂ O ₃	19.2/21.3	93/99	-3.7/-3	0.6	1.1×10 ⁻⁵	1.2×10 ⁷
(PEA) ₂ (MA) ₂ Pb ₃ I ₁₀	1.2/0.8	220/182	0.2/0	0.2	1.3×10 ⁻⁷	1.9×10 ³
Al ₂ O ₃ /(PEA) ₂ (MA) ₂ Pb ₃ I ₁₀	20.4/20.5	82/80	-3.8/-3.9	0.1	1.5×10 ⁻⁵	1.8×10 ⁷

F: Forward scan B: Backward scan

Table S1. Comparison in device performance with different dielectric.

6. What could be the reason for a high gamma value of 2.3?

The high gamma value of 2.3 represents that the photoresponsivity increases with increased incident power, which is similar to other reported photodetector⁶. Under a strong light excitation, photon penetration depth increases, and part of photo-generated holes can drift to Al₂O₃/2D perovskite interface instead of recombination, leading to increased photoresponsivity.

7. The energy band diagram in Fig. 4a under $V_g < V_{Rc}$ does not seem to be correct. As per Fig. 3a the transistor is below threshold (close to being off) at large negative V_g . In this case the band bending should be in the opposite direction than what is shown, and MoS₂ is depleted (not accumulated as shown). The electric field in the perovskite will be in the opposite direction leading to electron accumulation at Al₂O₃/perovskite interface.

We appreciate for the reminder. We have corrected the energy band diagram in Fig. 4a. In view of the natural energy band alignment resulting from the varying 2D perovskite phases (Fig. 1e in the manuscript), the 2D perovskite band could bend upward at Al₂O₃/2D perovskite interface under $V_g < V_{Rc}$. Thus, the photo-generated holes can accumulate at Al₂O₃/2D perovskite interface and lead to photogating effect.

Figure 3. Energy band diagram of the device under a gate bias lower than V_{Rc} .

8. The explanation for VRC crossover is not clear- if it is due to increasing electric field in the perovskite, what determines the critical electrical field- corresponding to VRC, in terms of perovskite material parameters, where the light intensity dependence of R flips over? Further, if a built-in electric field due to hole accumulation opposes further hole drift to the interface (Fig. 4f), shouldn't the R value saturate for increasing light intensity?

We are sorry for the confusion. In view of the charge spatial separation throughout the vertical direction of the perovskite film, the V_{Rc} crossover is probably

determined by the band bending induced built-in electric field in 2D perovskite. In principle, the V_{Rc} value can be modulated through controlled composition variation in 2D perovskite. In addition, because the hole accumulation opposes further hole drift to $\text{Al}_2\text{O}_3/2\text{D}$ perovskite interface (Fig. 4f), the R value tends to saturate as the increase of the laser power density according to the equation, $R = I_{ph}/P_{light}S \sim P_{light}^{\gamma-1}$ ($\gamma < 1$).

9. Can the authors show stability of optical photoresponse over a period of time?

We appreciate for the reminder. The stability of the phototransistor with $\text{Al}_2\text{O}_3/2\text{D}$ perovskite dielectric is investigated. The photoresponsivity retain 98% and 89% of its original value after exposing in nitrogen and air environment for more than 160 hours, respectively. It is known that perovskites usually exhibit poor stability. Here, the remarkable stability of the device can be attributed to the hydrophobicity of the organic spacer in 2D perovskite^{7,8}, which shields the inner perovskite layers from moisture.

Figure 4. Stability measurements of the phototransistor with $\text{Al}_2\text{O}_3/2\text{D}$ perovskite dielectric at $P_{light} = 1 \text{ mW/cm}^2$ and $\lambda = 914 \text{ nm}$.

10. There are a large number of writing errors, the manuscript needs thorough proof reading. A few examples:

- (a) line 206 – “stack” should be “stacked”
- (b) line 168- “charges” should “charge”
- (c) line 92 – “consisted” should be “consisting”

We thank for the reminder and sorry for the mistakes. We have corrected them and revised the manuscript carefully.

Overall, we thank the referee for the detailed comments and suggestions, which help to greatly improve our manuscript. We believe the photoactive dielectric presented here will make significant contribute to high-performance perovskite-based photodetection for practical utilizations.

References

1. Fiori, G. *et al.* Electronics based on two-dimensional materials. *Nat. Nanotechnol.* **9**, 768-779 (2014).
2. Sarker, B. K. *et al.* Position-dependent and millimetre-range photodetection in phototransistors with micrometre-scale graphene on SiC. *Nat. Nanotechnol.* **12**, 668 (2017).
3. Huo, N., Gupta, S., Konstantatos, G., Gupta, S. & Konstantatos, G. MoS₂-HgTe quantum dot hybrid photodetectors beyond 2 μm . *Adv. Mater.* **29**, 1606576 (2017).
4. Juarez-Perez, E. J. *et al.* Photoinduced giant dielectric constant in lead halide perovskite solar cells. *J. Phys. Chem. Lett.* **5**, 2390-2394 (2014).
5. Hirai, H., Tsuchiya, H., Kamakura, Y., Mori, N. & Ogawa, M. Electron mobility calculation for graphene on substrates. *J. Appl. Phys.* **116**, 083703 (2014).
6. Guo, F. *et al.* A nanocomposite ultraviolet photodetector based on interfacial trap-controlled charge injection. *Nat. Nanotechnol.* **7**, 798-802 (2012).
7. Shao, Y. *et al.* Stable graphene-two-dimensional multiphase perovskite heterostructure phototransistors with high gain. *Nano Lett.* **17**, 7330-7338 (2017).
8. Jian, Q. *et al.* Aligned and graded type-II Ruddlesden–Popper Perovskite films for efficient solar cells. *Adv. Energy Mater.* **8**, 1800185 (2018).

Response to Reviewer # 2:

We thank the referee for careful reading the manuscript and providing a number of precious comments. And we have addressed the comments carefully with listed response below. Meanwhile we have revised our manuscript accordingly.

In this work, the authors present a simple yet universal $Al_2O_3/2D$ perovskite heterostructure dielectric for high-performance phototransistors. The type-II energy band alignment in 2D perovskite facilitates the effective spatial separation of photoexcited carriers, thus inducing the outstanding photoresponse properties. By adding a high-k Al_2O_3 layer between the 2D perovskite layer and the semiconductor layer, the two effects of charge trapping and remnant polarization under gate bias are found to be neutralized with each other, resulting in low-voltage transistors of negligible hysteresis and improved reliability. The devices also exhibit excellent linear dynamic characteristics.

Overall, I think this paper is sufficiently innovative and the experiments are nicely executed. I think the work will be of interest to diverse researchers working on perovskite materials and 2D-material phototransistors and I would recommend its publication in Nature Communications after the following questions are clarified.

1. Although there has been significant progress on the development of perovskite materials in recent years, the poor stability is still a grand challenge yet to be tackled. How's the stability of the devices reported in this paper and what are the possible approaches to improve the lifetime of perovskite-based devices? Some discussions would be helpful.

We thank for the referee's detailed comments on these viewpoints. Although a huge number of articles about perovskites have been reported, improving stability of perovskites is still a grand challenge yet to be tackled. There are three main intrinsic factors leading to perovskite instability: hygroscopicity, thermal instability, and ionic migration¹. In previous studies, several strategies have been proposed to improve device stability. The hygroscopicity can be solved by encapsulation². The thermal instability can be addressed by composition tuning to increase the decomposition energy or barrier³. Lastly, the issue of ionic migration is currently treated by A site alkali doping and replacement^{4,5}, multiple dimensional perovskites engineering⁶, and organic molecular additives⁷. Therefore, we are optimistic that perovskites would perform extraordinarily well outside in an unshaded location for a long time. In our experiment, the photoresponsivity retain 98% and 89% of its original values after exposing in nitrogen and air environment for more than 160 hours, respectively. The remarkable stability of the device can be attributed to the hydrophobicity of the organic spacer^{8,9}, which shields the inner perovskite layers from moisture. We have added the relevant discussion in the manuscript.

Figure 1. Stability measurements of the phototransistor with $\text{Al}_2\text{O}_3/2\text{D}$ perovskite dielectric at $P_{\text{light}} = 1 \text{ mW/cm}^2$ and $\lambda = 914 \text{ nm}$.

2. The device is capable of broadband photodetection ranging from 457 nm to 1064 nm. However, the absorption edge of the upper perovskite layer is located at around 800 nm. Is the MoS_2 layer responsible for the near-infrared response?

We thank for the reminder. As shown in Fig. 1f, the near-infrared photoresponsivity of the pristine MoS_2 device is several orders of magnitude less than the $\text{MoS}_2/\text{Al}_2\text{O}_3/2\text{D}$ perovskite device. Therefore, the impressive near-infrared response is mainly caused by the 2D perovskite layer. This is probably due to the efficient excitation of carriers from the valence band to the trap states within the perovskite bandgap, being similar to the extrinsic photoconductors based on conventional semiconductors¹⁰.

3. Page 12, Line 5, the authors estimated the $I_{\text{light}}/I_{\text{dark}}$ ratio to be 2.6×10^6 at $V_{\text{gs}} = -3.5 \text{ V}$ where the incident power density (P_{light}) is $\sim 1 \text{ mW/cm}^2$. The electrical measurements were carried out by varying the intensity from $\sim 0.01 - \sim 1 \text{ mW/cm}^2$. In this case how can the author say that P_{light} is as low as 1 mW/cm^2 ?

We thank for the referee's detailed comments. The $I_{\text{light}}/I_{\text{dark}}$ ratio is a typical parameter employed to evaluate the degree of obtrusive noise. Generally, the $I_{\text{light}}/I_{\text{dark}}$ ratio increases with increased incident power density. In comparison with previously reported photodetectors, the P_{light} of 1 mW/cm^2 is a relatively low value¹¹. The high $I_{\text{light}}/I_{\text{dark}}$ ratio of 2.6×10^6 arises from the profound photogating effect in our device.

4. What are the channel length and width of the phototransistors?

We thank for the reminder. The channel length is $3 \mu\text{m}$, and the channel width is $3\text{-}10 \mu\text{m}$. We have added the relevant description in the manuscript.

5. First of all, Fig. 2b has not been mentioned in the manuscript. Furthermore, I assumed the arrows in Fig. 2b is for double sweep to show hysteresis (or lack thereof) in the I_{ds} - V_{ds} characteristics of the phototransistors. Usually people present hysteresis on transfer I_{ds} - V_{gs} characteristics of the devices (e.g. Fig. 2a of the paper). It is not clear to me why it is necessary to also present it for the I_{ds} - V_{ds} characteristics.

We thank for the reminder. We have added the relevant description in the manuscript. The output curves in Fig. 2b indicate the good ohmic contacts between the Cr/Au electrodes and the MoS₂. In addition, the trapping of electrons at MoS₂/Al₂O₃ interface or migration of ionic vacancies in 2D perovskite can also induce the hysteresis phenomenon during I_{ds} - V_{ds} characteristic measurement. Here, the negligible hysteresis indicates the operational stability of the device.

Overall, we thank the referee for the detailed comments and suggestions, which help to greatly improve our manuscript. We believe the photoactive dielectric presented here will make significant contribute to high-performance perovskite-based photodetection for practical utilizations.

References

1. Meng, L. *et al.* Addressing the stability issue of perovskite solar cells for commercial applications. *Nat. Commun.* **9**, 1-4 (2018).
2. Wang, P. *et al.* Solvent-controlled growth of inorganic perovskite films in dry environment for efficient and stable solar cells. *Nat. Commun.* **9**, 1-7 (2018).
3. Arora, N. *et al.* Perovskite solar cells with CuSCN hole extraction layers yield stabilized efficiencies greater than 20%. *Science* **358**, 768-771 (2017).
4. Saliba, M. *et al.* Incorporation of rubidium cations into perovskite solar cells improves photovoltaic performance. *Science* **354**, 206-209 (2016).
5. Zhou, W. *et al.* Light-independent ionic transport in inorganic perovskite and ultrastable Cs-based perovskite solar cells. *J. Phys. Chem. Lett.* **8**, 4122-4128 (2017).
6. Tsai, H. *et al.* High-efficiency two-dimensional Ruddlesden-Popper perovskite solar cells. *Nature* **536**, 312-316 (2016).
7. Zong, Y. *et al.* Continuous grain-boundary functionalization for high-efficiency perovskite solar cells with exceptional stability. *Chem* **4**, 1404-1415 (2018).
8. Shao, Y. *et al.* Stable graphene-two-dimensional multiphase perovskite heterostructure phototransistors with high gain. *Nano Lett.* **17**, 7330-7338 (2017).
9. Jian, Q. *et al.* Aligned and graded type-II Ruddlesden-Popper Perovskite films for efficient solar cells. *Adv. Energy Mater.* **8**, 1800185 (2018).
10. Teitsworth, S. W. & Westervelt, R. M. Chaos and broadband noise in extrinsic photoconductors. *Phys. Rev. Lett.* **53**, 2587 (1984).
11. Koppens, F. H. L. *et al.* Photodetectors based on graphene, other two-dimensional materials and hybrid systems. *Nat. Nanotechnol.* **9**, 780 (2014).

Response to Reviewer # 3:

We thank the referee for careful reading the manuscript and providing a number of precious comments. And we have addressed the comments carefully with listed response below. Meanwhile we have revised our manuscript accordingly.

Jiang et al. have demonstrated the fabrication of a high performance MoS₂ phototransistor employing a novel Al₂O₃/2D perovskite hybrid dielectric configuration. The results highlight the achievement of both ultrahigh responsivity and excellent linear dynamic range in one device. In addition, this work on Al₂O₃/2D perovskite dielectric to achieve greatly improved operational stability is very crucial for the technology to be applied in practical fields. I appreciate the experimental work which is systematically performed and discussed. Considering that the overall quality and significance of this work, I would like to recommend this work for publication after the authors address the following questions:

1. In the Al₂O₃/2D perovskite hybrid dielectric, the thickness of Al₂O₃ layer is 9 nm. In my understanding, the Al₂O₃ layer is employed as a high-k dielectric, and the thinner thickness could help to enhance the gate capacitance. Does the Al₂O₃ thickness influence the performance of the device?

We thank for the referee's detailed comments on these viewpoints. In our device architecture, the 2D perovskite cover the Al₂O₃ layer completely. Although the thinner Al₂O₃ layer is beneficial for low-voltage phototransistor operation, it could induce an enlarged leakage current. Therefore, the thickness of Al₂O₃ layer in our experiments is optimized to 9 nm.

2. In Figure 3d, the photoresponsivity remains almost constant within the light power density ranging from 7.2 to 1232.2 $\mu\text{W}/\text{cm}^2$. What is the R-squared (coefficient of determination) of the linear fitting for the device?

We thank for the reminder. By linearly fitting the data, we find that the *R*-squared (coefficient of determination) of the linear fitting for the device is 0.998, which is very close to 1, indicating the perfect linearity of the data. We have added the relevant discussion in the manuscript.

Figure 1. The linear fitting result for the device.

3. Why can the devices detect near-infrared light? The perovskites are usually employed for visible light detection, which is in accordance with the absorption spectrum shown in Figure S1.

As shown in Fig. 1f, the near-infrared photoresponsivity of the pristine MoS₂ device is several orders of magnitude less than the MoS₂/Al₂O₃/2D perovskite device. Therefore, the impressive near-infrared photoresponse is mainly caused by the 2D perovskite layer, which is probably due to the efficient excitation of carriers from the valence band to the traps states within the bandgap, being similar to the extrinsic photoconductors based on conventional semiconductors¹.

4. Line 8, Page 3. the sentence “...is also become more eminent.” should be corrected; Line 7, Page 10. “...and S denotes the active area of photodetector, ...” Compared to the illumination area, which is larger in the experiment setup? The authors should specify this value in the manuscript.

We thank for the reminder and sorry for the mistake. We have corrected the description in the manuscript. In addition, the active area of photodetectors represents the area of MoS₂ channel, which is slightly smaller than the illumination area. We have added the relevant description in the manuscript.

5. Line 16-18, Page 12. “In general, the γ value is in the range of $0 < \gamma < 1$ for photogating devices, ...photoconductive effect.” The author should elaborate more on this point, as this value depends also on the surface defects and quality of film.

We thank for the referee’s reminder. We have corrected the description in the manuscript, “The nonunity exponent of $0 < \gamma < 1$ is often observed in photogating devices^{2,3}, as a result of the complex process of carrier generation, trapping, and recombination within semiconductors. The photocurrent tends to saturate as the increase of the laser power, which is partly due to the gradually filled trap states. The

smaller γ value represents the more prominent photogating effect, while $\gamma = 1$ represents the pure photoconductive effect.”

6. In Figure 2a and Figure 5a, what is the source-drain voltage used in the measurements?

We appreciate for the reminder. The source-drain voltage is set to 1 V. We have added the relevant description in the manuscript.

Overall, we thank the referee for the detailed comments and suggestions, which help to greatly improve our manuscript. We believe the photoactive dielectric presented here will make significant contribute to high-performance perovskite-based photodetection for practical utilizations.

References

1. Teitworth, S. W. & Westervelt, R. M. Chaos and broadband noise in extrinsic photoconductors. *Phys. Rev. Lett.* **53**, 2587 (1984).
2. Hu, C. *et al.* Synergistic effect of hybrid PbS quantum dots/2D-WSe₂ toward high performance and broadband phototransistors. *Adv. Funct. Mater.* **27**, 1603605 (2017).
3. Island, J. O., Blanter, S. I., Buscema, M., van der Zant, H. S., & Castellanos-Gomez, A. Gate controlled photocurrent generation mechanisms in high-gain In₂Se₃ phototransistors. *Nano lett.* **15**, 7853-7858 (2015).

REVIEWER COMMENTS

Reviewer #1 (Remarks to the Author):

Although the authors have made attempts to correct several errors in the manuscript, which is an indicator of how poorly the manuscript was written in the first place, the scientific aspects are still unconvincing and inconsistent. The authors need to go back to the drawing board, perform more analysis and do additional experiments to justify their proposed mechanism. Some examples below:

(i) The choice of MoS₂ is still not clear because the device exploits neither its optical properties nor any specific electronic property for this device. The authors claim that improved gate control can lead to better short-channel effects and low power operation but this is true for logic transistor technologies and not necessarily phototransistors. Also, the authors have used long gate lengths and 5 nm thick MoS₂, where short channel effects do not come into play. Which specific photodetector metric is being benefitted by the choice of MoS₂?

(ii) The authors have now corrected Fig. 1e, where electrons have been shown to be trapped in trap states in the 2D perovskite. But the authors haven't answered the question (in rebuttal or revised manuscript) of how these electrons move towards the gate electrode since they are trapped now.

(iii) Regarding the answer to the query on how the photogenerated holes in 2D perovskite recombine with supplemental electrons in MoS₂ the authors have speculated "In addition, the carrier lifetime is most likely determined by trap-assisted recombination through the traps or defect levels in Al₂O₃ or perovskite layer²⁷."

Recombination in Al₂O₃ is highly unlikely- the carriers are separated by a 9 nm thick dielectric, the conduction band offset for MoS₂/Al₂O₃ and valence band offset for 2D perovskite/Al₂O₃ – both are large (Fig. 1e) thereby impeding tunneling into Al₂O₃.

Similarly recombination in perovskite is also unlikely since electrons will have to tunnel from MoS₂ to perovskite through 9 nm Al₂O₃.

(iv) The authors mention in their rebuttal letter that "it is hard to compare the hysteresis due to perovskite ($n = 3$) alone and due to Al₂O₃ alone." But it is a central claim on the paper that "the two effects of charge trapping and ionic polarization under gate bias could be neutralized with each other, resulting in negligible hysteresis in MoS₂ phototransistor." A mathematical analysis of how this is achieved, is critical to support this claim. For instance, if it was a simple subtractive/additive process, can the authors show that an MoS₂/perovskite ($n=3$)/Al₂O₃/gate metal stack also has zero hysteresis?

(v) The authors' analysis of series capacitance effect (response to question 5) and V_{th} does not seem to be correct.

(a) If the 2D perovskite had a large low frequency dielectric constant then the "only $n=3$ " should have better sub-threshold characteristics than "only Al₂O₃" instead of 220 mV/decade vs 93 mV/decade

Also, irrespective of how high the dielectric constant is, SS cannot improve from 93 to 82 mv/dec by adding Al₂O₃ in series.

(b) The V_{th} of Al₂O₃ and Al₂O₃/PEA ($n=3$) are different by nearly a Volt in the figure below (see pdf copy being attached). How did the authors extract values of -3.7 and -3.8 V?

(vi) The authors have tried to correct the band diagram in Fig. 4a for MoS₂ based on my inputs. But, to stay consistent with their mechanisms, they had to keep the band bending direction the same in the 2D perovskite, leading to a discontinuity in electric field (hence new questions on interface charge etc.) at the Al₂O₃/perovskite interface. This has led to them to come up with highly speculative reasons (such as 2D perovskite phases) to justify the band bending in the perovskite.

This is a glaring example of poorly understood device physics and mechanisms, where principles of basic semiconductor textbook physics seem to have been glossed over.

Reviewer #2 (Remarks to the Author):

The authors have addressed all my previous comments in this revision. I recommend its publication in Nature Communications.

Reviewer #3 (Remarks to the Author):

The authors have addressed my comments carefully. The paper can be published now.

Response to Reviewer # 1:

We thank the referee for careful reading the manuscript and providing a number of precious comments. And we have addressed the comments carefully with listed response below. Meanwhile we have revised our manuscript accordingly.

Although the authors have made attempts to correct several errors in the manuscript, which is an indicator of how poorly the manuscript was written in the first place, the scientific aspects are still unconvincing and inconsistent. The authors need to go back to the drawing board, perform more analysis and do additional experiments to justify their proposed mechanism. Some examples below:

(i) The choice of MoS₂ is still not clear because the device exploits neither its optical properties nor any specific electronic property for this device. The authors claim that improved gate control can lead to better short-channel effects and low power operation but this is true for logic transistor technologies and not necessarily phototransistors. Also, the authors have used long gate lengths and 5 nm thick MoS₂, where short channel effects do not come into play. Which specific photodetector metric is being benefitted by the choice of MoS₂?

We thank for the referee's detailed comments on these viewpoints. High-performance photodetection in this device architecture relies strongly on Al₂O₃/2D perovskite heterojunction dielectric to efficiently separate photo-generated carriers and produce pronounced photogating effect, as well as a high-mobility conducting channel to generate electrical output. The choice of MoS₂ is due to its advantages in flexibility¹, ease of processing², high carrier mobility³, and especially ultimate scaling opportunity⁴. In principle, other low-dimensional materials and conventional semiconductors can also be employed as the conducting channel. We will explore the possibility in future work.

(ii) The authors have now corrected Fig. 1e, where electrons have been shown to be trapped in trap states in the 2D perovskite. But the authors haven't answered the question (in rebuttal or revised manuscript) of how these electrons move towards the gate electrode since they are trapped now.

We thank for the referee's detailed comments and sorry for this confusion. The trapped electrons are difficult to move towards the gate electrode. While the

photo-generated electrons are trapped in 2D perovskite, the photo-generated holes can move towards Al_2O_3 /2D perovskite interface. The spatial separation of the photo-generated carriers in 2D perovskite would lead to an increase in the electric field at the MoS_2 and thus the conductivity⁵.

(iii) Regarding the answer to the query on how the photogenerated holes in 2D perovskite recombine with supplemental electrons in MoS_2 the authors have speculated “In addition, the carrier lifetime is most likely determined by trap-assisted recombination through the traps or defect levels in Al_2O_3 or perovskite layer²⁷.” Recombination in Al_2O_3 is highly unlikely- the carriers are separated by a 9 nm thick dielectric, the conduction band offset for $\text{MoS}_2/\text{Al}_2\text{O}_3$ and valence band offset for 2D perovskite/ Al_2O_3 – both are large (Fig. 1e) thereby impeding tunneling into Al_2O_3 . Similarly recombination in perovskite is also unlikely since electrons will have to tunnel from MoS_2 to perovskite through 9 nm Al_2O_3 .

We thank for the reminder and sorry for this mistake. The photo-generated holes in 2D perovskite can not recombine with the supplemental electrons in MoS_2 . The answer to the query is not right. The reasonable speculate is that the photo-generated holes gradually recombine with the opposite charge in 2D perovskite after the laser is switched off. Because the additional electrons in MoS_2 are induced by the accumulation of photo-generated holes at the Al_2O_3 /2D perovskite interface, the electrons in MoS_2 would eventually return to the pre-illumination levels. We have corrected the relevant description in revised manuscript.

(iv) The authors mention in their rebuttal letter that “it is hard to compare the hysteresis due to perovskite ($n = 3$) alone and due to Al_2O_3 alone.” But it is a central claim on the paper that “the two effects of charge trapping and ionic polarization under gate bias could be neutralized with each other, resulting in negligible hysteresis in MoS_2 phototransistor.” A mathematical analysis of how this is achieved, is critical to support this claim. For instance, if it was a simple subtractive/additive process, can the authors show that an MoS_2 /perovskite ($n=3$)/ Al_2O_3 /gate metal stack also has zero hysteresis?

We thank for the referee’s detailed comments and sorry for the confusion. The

2D perovskite/ Al_2O_3 stack cannot be fabricated due to the decomposition of 2D perovskite, as a result of reaction with H_2O precursor during ALD process. The hysteresis elimination is based on a balance between two contrasting effects i.e., the charge trapping in the Al_2O_3 layer and the ionic migration in the 2D perovskite layer. We quantitatively analyzed the stretched exponential decay of the drain current in the manuscript. The result indicates the combined effects of charge trapping and ionic migration on improving the device stability.

(v) The authors' analysis of series capacitance effect (response to question 5) and V_{th} does not seem to be correct.

(a) If the 2D perovskite had a large low frequency dielectric constant then the "only $n=3$ " should have better sub-threshold characteristics than "only Al_2O_3 " instead of 220 mV/decade vs 93 mV/decade

Also, irrespective of how high the dielectric constant is, SS cannot improve from 93 to 82 mv/dec by adding Al_2O_3 in series.

(b) The V_{th} of Al_2O_3 and $\text{Al}_2\text{O}_3/\text{PEA}$ ($n=3$) are different by nearly a Volt in the figure below (see pdf copy being attached). How did the authors extract values of -3.7 and -3.8 V?

(a) We thank for the referee's detailed comments. Besides the gate capacitance, the interface state density at the channel/gate insulator interface also affects the subthreshold swing. In comparison with the Al_2O_3 dielectric, the larger SS value using 2D perovskite dielectric can be attributed to the worse channel/insulator interface. In addition, the slight difference in SS values (93 versus 82) for the two devices is probably caused by the unsatisfactory uniformity of the MoS_2 flake.

(b) We thank for the reminder. The V_{th} values are extracted from the Fig. S4b in the supplementary information instead of Fig. 2a in the manuscript. In the previous version, the V_{th} values are determined from the horizontal intercept of a linear part in $I_{ds}^{1/2}$ versus V_{gs} plot. However, in view of the linear region operation of MoS_2 transistor, the V_{th} value should be estimated from the I_{ds} versus V_{gs} plot instead of square root of I_{ds} versus V_{gs} (as shown in Figure 1). We have corrected the V_{th} values in the Table S1.

Figure 1. Transfer characteristics of MoS₂ phototransistors with three types of dielectrics. The horizontal intercept of a linear part in I_{ds} versus V_{gs} plot is employed to estimate the V_{th} value with **a** Al₂O₃, **b** (PEA)₂(MA)₂Pb₃I₁₀, and **c** Al₂O₃/(PEA)₂(MA)₂Pb₃I₁₀ dielectrics, respectively.

Dielectric structure	μ_F/μ_B (cm ² /V·s)	SS _F /SS _B (mV/dec)	V_{thF}/V_{thB} (V)	Hysteresis (V)	I_{on} (A/ μ m)	I_{on}/I_{off}
Al ₂ O ₃	19.2/21.3	93/99	-3.2/-2.5	0.7	1.1×10^{-5}	2.8×10^8
(PEA) ₂ (MA) ₂ Pb ₃ I ₁₀	1.2/0.8	220/182	0.3/0.1	0.2	1.3×10^{-7}	1.9×10^3
Al ₂ O ₃ /(PEA) ₂ (MA) ₂ Pb ₃ I ₁₀	20.4/20.5	82/80	-3.2/-3.3	0.1	1.5×10^{-5}	4.8×10^8

F: Forward scan B: Backward scan

Table S1. Comparison in device performance with different dielectrics.

(vi) The authors have tried to correct the band diagram in Fig. 4a for MoS₂ based on my inputs. But, to stay consistent with their mechanisms, they had to keep the band bending direction the same in the 2D perovskite, leading to a discontinuity in electric field (hence new questions on interface charge etc.) at the Al₂O₃/perovskite interface. This has led to them to come up with highly speculative reasons (such as 2D perovskite phases) to justify the band bending in the perovskite.

We thank for the referee's detailed comments and sorry for this mistake. In the band diagram shown in Fig. 4a in the manuscript, the gate bias is below threshold but higher than the voltage where the device is in off state. At this point, the diffusion current in MoS₂ is caused by gradient of electron concentration. Therefore, the band diagram for MoS₂ in the previous version was correct. We have replaced it with the initial version (Figure 2). In order to avoid confusion, we have changed the description "a Energy band diagram of the device under a gate bias lower than V_{Rc} " into "a Energy band diagram of the device under a gate bias below V_{Rc} but higher than

the voltage where the device is in off state” in revised manuscript.

Figure 2. Energy band diagram of the device under a gate bias below V_{Rc} but higher than the voltage where the device is in off state.

References

1. Manzeli, S., Ovchinnikov, D., Pasquier, D., Yazyev, O. V. & Kis, A. 2D transition metal dichalcogenides. *Nat. Rev. Mater.* **2**, 17033 (2017).
2. Lee, Y. H. *et al.* Synthesis of large-area MoS₂ atomic layers with chemical vapor deposition. *Adv. Mater.* **24**, 2320-2325 (2012).
3. Desai, S. B. *et al.* MoS₂ transistors with 1-nanometer gate lengths. *Science* **354**, 99-102 (2016).
4. Fiori, G. *et al.* Electronics based on two-dimensional materials. *Nat. Nanotechnol.* **9**, 768-779 (2014).
5. Sarker, B. K. *et al.* Position-dependent and millimetre-range photodetection in phototransistors with micrometre-scale graphene on SiC. *Nat. Nanotechnol.* **12**, 668 (2017).

REVIEWERS' COMMENTS:

Reviewer #1 (Remarks to the Author):

The authors have revised the manuscript appropriately and answered all questions satisfactorily.

Reviewer #2 (Remarks to the Author):

The authors have addressed all my comments. I recommend the publication in Nature Communications in its present form.

Reviewer #3 (Remarks to the Author):

The authors have addressed the comments raised by reviewers carefully and adequately, and I recommend the publication of this paper in Nature Communications.